# Massa Medicata Fermentata, a Functional Food for Improving the Metabolic Profile via Prominent Anti-Oxidative and Anti-Inflammatory Effects

**DOI:** 10.3390/antiox13101271

**Published:** 2024-10-19

**Authors:** Kyung-Mi Jung, Ga-Ram Yu, Da-Hoon Kim, Dong-Woo Lim, Won-Hwan Park

**Affiliations:** 1Department of Diagnostics, College of Korean Medicine, Dongguk University, Goyang-si 10326, Republic of Korea; chaoskmj@hanmail.net (K.-M.J.); kalama2@dongguk.edu (G.-R.Y.); lilwayne224@gmail.com (D.-H.K.); 2Institute of Korean Medicine, Dongguk University, Goyang-si 10326, Republic of Korea

**Keywords:** anti-inflammation, anti-oxidant, metabolism, Massa Medicata Fermentata, nutraceutical, functional food

## Abstract

Massa Medicata Fermentata (MMF) is a naturally fermented product used to treat indigestion and increase stomach activity in traditional medicine. This study examined the ability of the hydrothermal extract of MMF to scavenge free radicals corresponding to biological oxidative stresses, further protecting essential biomolecules. The anti-inflammatory effects of MMF were evaluated in LPS-induced RAW264.7 macrophages and zebrafish. In addition, the effects of MMF on the body mass index (BMI) and cholesterol accumulation in adult zebrafish fed a high-cholesterol diet (HCD) for three weeks were examined. MMF prevented the DNA and lipid damage caused by oxidative stress, inhibited LDL oxidation, and reduced the expression of cytokines and related proteins (MAPK and NFκB), with prominent anti-oxidative pathway (NRF2-HO-1) activation properties. LPS-induced NO production was reduced, and the increase in BMI and TC caused by the HCD diet was suppressed by MMF in zebrafish embryos or adult zebrafish. The bioactive aglycone of quercetin may be contributing to the mechanisms of systemic effects. MMF has excellent antioxidant properties and is useful for improving inflammation status and metabolic profile, thus highlighting its potential as a healthy, functional food.

## 1. Introduction

The interest in health-functional foods, such as nutritional supplements, herbal medicines, and health supplements, has increased steadily. They help maintain health, prevent disease, and improve the quality of life [1,2]. The health benefits of a plant-based diet are related to the antioxidant and anti-inflammatory mechanisms of various phytochemicals [3]. Recent studies have shown that both obesity and hypertension are associated with oxidative stress and inflammatory response. Oxidative stress and inflammatory processes adversely affect endothelial dysfunction in cardiovascular diseases and induce insulin resistance [4,5]. Insulin resistance plays a crucial role in the development and progression of metabolic diseases such as hypertension, diabetes, and non-alcoholic fatty liver disease [6].

The excessive production of reactive oxygen species (ROS), including superoxide radical anions, hydroxyl radicals, singlet oxygen, and hydrogen peroxide, induces oxidative stress and protein oxidation, contributing to the pathogenesis of various types of inflammatory diseases, such as atherosclerosis, rheumatoid arthritis, cancer, and allergies [7,8,9]. Phytochemicals with antioxidant effects have been studied extensively to treat inflammatory diseases [10]. Nrf2 (nuclear factor erythroid 2-related factor 2) is a major cellular sensor of oxidative stress. Under normal conditions, Nrf2 binds to Kelch-like ECH-associated protein 1 (Keap1) in the cytoplasm [11]. When the cells are exposed to oxidative stress, Nrf2 translocates to the nucleus. It binds to and activates antioxidant response elements (AREs), cis-acting enhancers in the promoter regions of a large and unique set of target genes, restoring redox homeostasis [12]. HO-1 (heme oxygenase 1) protects cells exposed to oxidizing agents. Increased HO-1 expression, which is regulated primarily by the Nrf2-ARE pathway, is associated with defense against oxidative stress [13,14,15].

Macrophages are immune cells that regulate the inflammatory process by releasing mediators such as cytokines and NO (nitric oxide). iNOS (inducible nitric oxide synthase) is a pro-inflammatory mediator that is key to assessing inflammation [16]. NO is generated in macrophages by iNOS following exposure to pro-inflammatory cytokines, including TNF-α, IL-1β, IL-6, and INF-γ, or microbial products, such as LPS [17,18]. The continuous and excessive production of NO can lead to a variety of diseases, such as atherosclerosis, diabetes, obesity, and cancer. The activation of NFκB (nuclear factor kappa-light-chain-enhancer of activated B cells) and MAPKs (mitogen-activated protein kinases) family members, including ERK (extracellular signal-regulated kinase), JNK (c-Jun N-terminal kinase), and p-p38 (p38 mitogen-activated protein kinase), are important for the production of pro-inflammatory factors and are triggered during inflammation under LPS induction in macrophages [19]. Thus, suppressing the NFκB and MAPK signaling pathways is a pharmacological approach to diseases such as cancer, arthritis, Alzheimer’s disease, asthma allergies, and other inflammatory diseases [20,21].

MMF (Massa Medicata Fermentata) is used in oriental medicine to treat gastrointestinal disorders, such as indigestion, because it strengthens the spleen and stomach and promotes digestion [22,23]. In addition, MMF is widely used to treat vomiting, diarrhea, obesity, and related metabolic diseases. MMF is a natural fermentation product of bitter almond (the kernel of *Prunus armeniaca* L.), red bean (*Vigna umbellata* Thunb.), *Xanthium sibiricum* L., *Polygonum hydropiper* L., *Artemisia annua* L., and wheat bran or flour (*Triticum aestivum* L.) combined in a certain proportion and fermented by microorganisms [24,25,26].

Fermentation is a feasible strategy for improving the efficacy of natural products, which are widely used in food and drug processing [23,27]. For example, in Korean medicine, the raw materials of Samjunghwan, a traditional prescription, are subjected to a long-term fermentation process to improve bioactivity [27,28]. The compounds from natural products are metabolized into smaller ones, showing better absorption and bioactivities [29].

As previously reported, the composition of MMF changes during the fermentation process [30]. Many fermented herbal medicines were reported to benefit health, ultimately leading to improved metabolic profiles with multifaceted mechanisms [31,32]. Therefore, the MMF can be used as food or medicine to treat metabolic diseases.

This study evaluated the overall efficacy and mechanisms of MMF for combating the exogenous oxidative and endogenous inflammatory factors. Furthermore, based on these general activities, the therapeutic effects of MMF on imbalanced metabolic profiles in HCD-fed zebrafish were investigated to evaluate its potential as a functional food.

## 2. Materials and Methods

### 2.1. Chemicals

Fetal bovine serum (FBS) and penicillin/streptomycin solution were purchased from Invitrogen (Carlsbad, CA, USA), Dulbecco’s Modified Eagle’s Medium (DMEM), and Dulbecco’s phosphate buffered saline (DPBS) were obtained from Gibco (Carlsbad, CA, USA). Lipopolysaccharide (LPS), 2,2-diphenyl-1-picrylhydrazyl (DPPH), human low-density lipoprotein (LDL), pBR322 plasmid DNA, DAF-FM DA (4-amino-5-methylamino-2′,7′-difluorofluorescein diacetate), and other reagents were acquired from Sigma Chemicals (St. Louis, MO, USA). The ELISA kits for mouse TNF-α, IL-1β, and IL-6 were supplied by R&D Systems (Minneapolis, MN, USA). The primary antibodies for iNOS (inducible nitric synthase), COX-2 (cyclooxygenase-2), Nrf2, HO-1, β-actin, and secondary antibodies were procured from Santa Cruz (Dallas, TX, USA). The primary antibodies against p-ERK, p-JNK, p-p38, p-IκB-α, and p-NF-κB (p-p65) were obtained from Cell Signaling Technology (Beverly, MA, USA). The total cholesterol (TC) assay kit was acquired from Biomax (Guri, Republic of Korea).

### 2.2. Preparation of MMF Extract

MMF was purchased from Humanherb (Chungcheongbuk-do, Republic of Korea). MMF (200 g) was ground to a powder and extracted in hot water (1 L) for 1 h at 95 °C. The crude extracts were filtered through Whatman filter paper, concentrated using a rotary evaporator (Buchi, Switzerland) at 95 °C, and freeze-dried to obtain a powder (12.5 g), which was eluted with DPBS and filtered through a 0.22 μm syringe filter before use.

### 2.3. Total Phenolic Content of MMF

The total phenolic content (TPC) was estimated using the Folin–Ciocalteu colorimetric method described previously with a slight modification. Briefly, 40 μL of MMF (10 mg/mL) was mixed with 200 μL of Folin–Ciocalteu reactant and 1.16 mL of distilled water and reacted for 3 min at room temperature and neutralized by adding 600 μL of 2% Na_2_CO_3_. The optical densities (ODs) of the reactants were measured at 765 nm using a microplate reader (Versamax, Molecular Devices, Silicon Valley, CA, USA) after incubation for 30 min [33]. Quantification was performed based on the standard curve of gallic acid. The results are expressed as grams of gallic acid equivalent (GAE) per 1 g of dry weight (DW). The total tannin content (TTC) was estimated using the Folin–Denis colorimetric method described previously, with a slight modification. Briefly, 15 μL of MMF (10 mg/mL) was mixed with 8.485 mL of distilled water, 0.5 mL of Folin–Denis reagent, and 1 mL of Na_2_CO_3_ and reacted at room temperature. The optical densities (ODs) of the reactants were measured at 700 nm using a microplate reader (Versamax, Molecular Devices, USA) [34]. Quantification was conducted based on the standard curve of tannic acid. The results were expressed as grams of tannic acid equivalent (TAE) per 1 g of dry weight (DW). The total flavonoid contents (TFC) were measured using a modified aluminum chloride colorimetric method. Briefly, 250 μL of MMF (10 mg/mL) was mixed with 1 mL of distilled water and 75 μL of NaNO_2_ and reacted for 6 min at room temperature. Subsequently, 300 µL of 10% AlCl_3_ (*w*/*v*) was added to the mixture. After six minutes, 500 µL of 1 M NaOH was added. The optical densities (ODs) of the reactants were measured at 510 nm using a microplate reader (Versamax, Molecular Devices, USA) [35]. Quantification was conducted based on the standard curve of catechin. The results are expressed as grams of catechin equivalent (CE) per 1 g dry weight (DW).

### 2.4. DPPH, ABTS Free Radical Scavenging Assay

The antioxidant capacity of the MMF, expressed as the donation of an electron or a hydrogen atom to the 2,2′-diphenyl-1-picrylhydrazyl (DPPH) free radical, was measured using a spectrophotometric method [36]. Briefly, 50 μL of MMF at various concentrations (50–800 μg/mL) was added to 1 mL of 0.1 mM DPPH in ethanol solution and 0.45 mL of 50 mM Tris-HCl (pH 7.4). The test tube was incubated in the dark for approximately 30 min at room temperature. Ascorbic acid was used as a positive control. The ODs of the reactants were measured at 517 nm using a microplate reader (Versamax, Molecular Devices, USA).
DPPH inhibition (%) = (A_B_ − A_C_)/A_B_ × 100
where A_B_ is the recorded absorbance of the blank solution, and A_C_ is the recorded absorbance of the MMF solution.

On a similar basis, the free radical scavenging activity of MMF was measured with 2,2′-Azino-bis(3-ethylbenzothiazoline-6-sulfonic acid) (ABTS) free radical scavenging assay. Briefly, a preincubate mixture of 7.4 mM ABTS and 2.6 mM potassium persulfate was kept for 24 h in the dark. The reaction buffer was mixed with samples in a 10:1 ratio and reacted for 10 min. The optical density of the reactant was measured at 732 nm wavelength.

### 2.5. Superoxide-Free—Radical Scavenging Assay

This assay was performed to assess the antioxidant activity of MMF in scavenging superoxide free radicals. The superoxide radicals were generated in vitro by the hypoxanthine/xanthine oxidase system [37]. The scavenging potential of the extract was determined using the nitro-blue tetrazolium (NBT) reduction method. A reaction mixture was prepared using a 50 mM phosphate buffer (pH 7.5) containing EDTA (0.05 mM), hypoxanthine (0.2 mM), NBT (1 mM), aqueous or ethanolic extract (distilled water for the control), and xanthine oxidase. The xanthine oxidase (1.2 U/μL) was added last. The ODs of the reactants were measured at 560 nm using a microplate reader (Versamax, Molecular Devices, USA).

### 2.6. Ferric Thiocyanate (FTC) Assay

The level of lipid peroxidation inhibition was examined by the ferric thiocyanate (FTC) method [38,39]. A mixture of 0.4 mL MMF in ethanol, 0.2 mL of 2.5% linolenic acid in absolute ethanol, and 0.4 mL of 50 mM phosphate buffer (pH 7.0) was placed in a vial and placed in an oven at 40 °C in the dark. A 0.1 mL sample of this solution was added to 3 mL of 70% ethanol and 0.01 mL of aqueous ammonium thiocyanate (30%, *w*/*v*) and kept for three minutes after adding 0.01 mL of ferrous chloride (2.45 mg/mL in 3.5% HCl) to the reaction mixture. The ODs of the reactants were measured at 500 nm using a microplate reader (Versamax, Molecular Devices, USA).

### 2.7. DNA Nicking Assay

The DNA nicking assay was performed as described [40]. A 9 μL sample of MMF (5.5 mg/mL) was added to 1 μL of supercoiled pBR322 plasmid DNA (0.5 μL). The mixture solution was incubated at room temperature for 10 min, followed by the addition of 10 μL of Fenton’s reagent (80 μmol/L FeCl_3_, 50 μmol/L ascorbic acid, and 30 mmol/L H_2_O_2_). The mixture solution was incubated at 37 °C for 5 min. Finally, the DNA was analyzed on 1.0% agarose gel electrophoresis (80 V, 30 min) and visualized under an ultraviolet illuminator. One unit of catalase reactant was used as the positive control.

### 2.8. Relative Electrophoretic Mobility (REM) Assay

The relative electrophoretic mobility (REM) of native low-density lipoprotein (LDL) and oxidized LDL was performed to evaluate the protective efficacy of MMF against LDL oxidation, according to the procedures of references [41]. Before electrophoresis, the samples containing 10 μM of CuSO_4_ and MMF were treated into LDL (120 μg/mL) and incubated for 12 h at 37 °C in the dark. The native LDL and pre-treated LDL samples loaded onto 0.7% agarose gel in TAE buffer (40 mM Tris-acetate with 1 mM EDTA, pH 8.0) and electrophoresed for 50 min at 85 V. After electrophoresis, the gel was fixed in 40% ethanol with 10% acetic acid for 30 min, stained with 0.15% coomassie brilliant blue R250, and LDL bands were visualized using the destaining solution.

### 2.9. Cell Culture and Treatment

RAW264.7 cells (a mouse macrophage cell line) were purchased from the Korea Cell Line Bank (KCLB, Seoul, Republic of Korea). The cells were cultured in DMEM medium supplemented with 10% FBS and 100 U/mL penicillin–streptomycin at 37 °C in a humidified atmosphere containing 5% CO_2_ and maintained at ~70% confluence before being used in the experiments. The cells were seeded on cell culture dishes at 1.2 × 10^5^ cells/mL and co-treated with LPS (1 μg/mL) and the MMF for 6–12 h (for immunofluorescence microscopy) or 12–24 h (for nitrite determination, Elisa, real-time PCR, and western blot).

### 2.10. Cell Viability Assay

The cell viabilities of RAW264.7 cells were determined by the WST (water-soluble tetrazolium salt) using an EZ-cytox assay kit according to the manufacturer’s instructions. Briefly, the cells were seeded on cell culture dishes at 1.2 × 10^5^ cells/mL and incubated at 37 °C in a humidified 5% CO_2_ incubator for 24 h. The cells were treated with various concentrations of MMF extract in the absence or presence of LPS (1 μg/mL) for 24 h. The medium was replaced with DMEM containing 10% EZ-cytox and reacted for 30 min. The ODs of the reactants were measured at 450 nm.

### 2.11. Nitrite Assay

Griess reagent was used to examine the effects of MMF on LPS-induced nitrite levels [42]. Briefly, the cells were seeded on cell culture dishes at 1.2 × 10^5^ cells/mL and incubated at 37 °C in a humidified 5% CO_2_ incubator for 24 h. The cells were co-treated with LPS (1 μg/mL) and MMF at various concentrations for 24 h. Supernatants were collected and mixed with Griess reagent. The ODs were measured at 570 nm. The nitrite concentrations were calculated using a standard curve.

### 2.12. Preparation of Nuclear and Cytoplasm Fractions

Nuclear and cytosolic proteins were separated using an extraction reagents kit from Thermo Fisher Scientific (Rockford, IL, USA) [43]. Briefly, RAW264.7 macrophages were seeded on cell culture dishes at 1.2 × 10^5^ cells/mL and incubated at 37 °C in a humidified 5% CO_2_ incubator for 24 h. The cells were then co-treated with LPS (1 μg/mL) and MMF at various concentrations for 6 h. The nuclear translocation of NFκB from the cytoplasm was assessed by western blot.

### 2.13. Western Blot Analysis

The levels of proteins associated with oxidation and inflammation were determined by western blot. Briefly, the cells were lysed with RIPA (radioimmunoprecipitation assay) buffer (Thermo Fisher Scientific, Rockford, IL, USA) containing a protease and phosphatase inhibitor cocktail (Gendepot, Barker, TX, USA). The protein concentrations were measured using the BCA (bicinchoninic acid) protein assay kit (Thermo Fisher Scientific, Rockford, IL, USA). Protein lysates (35 μg) were loaded into 7.5–10% SDS-PAGE gels, electrophoresed, and transferred to the PVDF membranes at 100 V for 60 min using an electrophoretic transfer cell (Bio-rad, Hercules, CA, USA). Membranes were then blocked with 5% BSA in TBS/T (TBS containing 0.1% Tween 20) for 2 h at room temperature, incubated with the primary antibodies (1:1500 dilution in TBS/T) overnight with gentle agitation, rinsed with TBS/T, and incubated with secondary antibodies (1:3000 dilution in TBS/T) at room temperature for 2 h. The blots were detected using the Fusion Solo imaging system (Vilber Lourmat, Collegien, France), and the proteins were visualized using an ECL buffer (Super Signal West Pico, Thermo Fisher Scientific).

### 2.14. Quantitative Real-Time Polymerase Chain Reaction

The mRNA expression levels of the pro-inflammatory mediators associated genes were determined by qPCR. The total mRNA was isolated using the Trizol reagent (Invitrogen, Carlsbad, CA, USA) according to the manufacturer’s instructions. Briefly, the isolated mRNA was subjected to cDNA synthesis using the AccuPower RT Premix kit (Bioneer, Daejeon, Republic of Korea) and oligo deoxythymine (dt) 18 primers (Invitrogen, Carlsbad, CA, USA). The primer-specific binding cDNA was amplified using a Light Cycler 480 PCR system (Roche, Basel, Switzerland). The PCR mix contained 10 μL of SYBR green master mixture (Roche, Switzerland), 8 μL of ultrapure water, 1 pmol/μL of gene primer, and 1 μL of template cDNA. Amplification was performed using the following schedule: initial denaturation at 95 °C for 10 min, followed by 45 cycles of denaturation at 95 °C for 10 s, annealing at 56–62 °C for 20 s, and extension at 72 °C for 20 s. Melting curve analysis was conducted at 95 °C for 5 min for a quality check. The threshold cycle (Ct) value was calculated to quantify the PCR results. The following primers were used: iNOS forward, 5′-GAGACAGGGAAGTCTGAAGCAC-3′, reverse, 5′-CCAGCAGTAGTTGCTCCTCTTC-3′; COX2 forward, 5′-GCGACATACTCAAGC AGGAGCA-3′, reverse, 5′-AGTGGTAACCGCTCAGGTGTTG-3′; TNFα forward, 5′-AAGCCTGTAGCCCACGTCGTA-3′, reverse, 5′-GGCACCACTAGTTGGTTGTCTTTG-3′; IL-1β forward, 5′-CTGAACTCAACTGTGAAATGCCA-3′, reverse, 5′-AAAGGTTTGGAAGCAGCCCT-3′; IL-6 forward, 5′-CCACTTCACAAGTCGGAGGCTTA-3′, reverse, 5′-GCAAGTGCATCATCGTTGTTCATAC-3′; β-actin forward, 5′-GCAAGTGCTTCTAGGCGGAC-3′, reverse, 5′-AAGAAAGGGTGTAAAACGCAGC-3′. The relative expression levels were calculated by dividing the gene Ct values by that of β-actin. All data were acquired using a LightCycler 480 instrument and software.

### 2.15. Enzyme-Linked Immunosorbent Assay (ELISA)

The concentrations of secretory inflammatory cytokines in cell culture supernatants were measured using Quantikine mouse ELISA kits (R&D Systems, Inc. Minneapolis, MN, USA). Briefly, the cells (1.2 × 10^5^ cells/mL) were seeded in six-well plates and incubated at 37 °C in a humidified atmosphere containing 5% CO_2_. Twenty-four hours after seeding, the cells were co-treated with LPS (1 μg/mL) and MMF for 24 h. The culture media were collected, and the TNFα, IL-1β, and IL-6 concentrations were measured according to the manufacturer’s instructions. The ODs were measured at 450 nm using a microplate spectrophotometer.

### 2.16. Immunofluorescence Microscopy

The nuclear translocation of Nrf2 and NFκB was followed by growing the cells on Lab-Tek II chamber slides (Nalge Nunc, Naperville, IL, USA) using a slight modification of the previous study [44]. Briefly, the cells were fixed in 4% formaldehyde for 5 min, permeabilized with 0.1% Triton X-100 for 10 min at room temperature, blocked with 1% BSA for 1 h at room temperature, and labeled with 2 μg/mL of the primary antibody overnight at 4 °C. NFκB in cytoplasm and nuclei was detected by treating the cells with 2 μg/mL of FITC containing 0.2% BSA for 1 h at room temperature. The nuclei were stained using a mounting medium containing API (Vector Laboratories, Newark, CA, USA). The images were captured under a fluorescence microscope (BX50, Olympus, Tokyo, Japan).

### 2.17. Zebrafish Maintenance

All zebrafish (AB strain) were maintained at 28 °C with a 14/10 h light/dark cycle. The embryos were harvested from natural mating and cultured in the embryo media. The fertilized embryos were incubated in 2 mg/L methylene blue containing the E3 embryo media (1.6 g KCl, 34.8 g NaCl, 5.8 g CaCl_2_·2H_2_O, and 9.78 g MgCl_2_·6H_2_O in 1 L of double-distilled water, pH 7.2 ± 0.1) at 28.5 °C. The effects of MMF on LPS-induced NO production in embryos were examined by stimulating the embryos with LPS (10 μg/mL) and MMF (500 μg/mL) for 24 h at 28.5 °C. After 7 h in the embryo medium E2 buffer, LPS-stimulated zebrafish larvae were transferred into 96-well plates and treated with a DAF-FM DA solution (2.5 μM) for 1 h in the dark at 28.5 °C, rinsed in fresh embryo medium, and fixed in 4% formaldehyde. The fluorescence intensities of individual larvae were assessed using a fluorescence microscope (BX50, Olympus, Japan). The effects of MMF on HCD-induced body mass index (BMI) and plasma total cholesterol in adult zebrafish were evaluated by selecting zebrafish with a standard length of approximately 3.8 cm and dividing them into four groups: control group; HCD 1 μg/mL group; HCD + MMF 10% group; and HCD + MMF 20% group. The HCD was prepared by mixing a diethyl ether solution of cholesterol with a normal feed to obtain 4 % (*w*/*w*) cholesterol in the diet after diethyl ether evaporation [45]. Each group was composed of ten fish and was fed an equal mass feed (30 mg/day, twice daily for 21 days) following the schedule, and the residual food was removed 1 h after feeding. The length and whole body weight of the zebrafish were measured. The BMI of the fish was obtained by dividing the body weight (g) of the fish by the square of the body length (cm). The zebrafish were sacrificed and analyzed individually, and the total cholesterol level was measured using commercial assay kits according to the manufacturer’s instructions.

### 2.18. HPLC Analyses

Chromatographic analyses of MMF were performed using an HPLC system (Agilent 1260 infinity HPLC, Agilent, Santa Clara, CA, USA) equipped with a UV-Diode array detector, degasser, and auto-sampler. Quercetin standard (Thermo Fisher Scientific, Rockford, IL, USA) was diluted with DMSO to 10 mM, and MMF extract was diluted with DMSO to 10 mg/mL. The main chemical components of fractionated samples were analyzed using an Agilent Eclipse XDB-C18 chromatographic column (150 × 4.6 mm, 5 µm pore size) at a flow rate of 1 mL/min, a column temperature of 23.0 °C, and a detection wavelength of 300 nm by Isocratic elution (mobile phase A—water with 0.1 formic acid and mobile phase B—acetonitrile with 0.1 formic acid) as follows: 1–20 min, 70% A:30% B.

### 2.19. Statistical Analyses

The data were analyzed using Graph Pad Prism version 5.0 software (Graph Pad, La Jolla, CA, USA). The standard curves were estimated using Excel 2010 and PowerPoint 2010 (Microsoft, Redmond, WA, USA). An analysis of the variance (ANOVA) and One-Way ANOVA with Dunnett’s multiple comparison test were used to determine the significance of the differences. The results are presented as means ± SDs, and *p*-values of <0.05 were considered significant.

## 3. Results

### 3.1. Total Phenolic Content and Radical Scavenging Activity

Figure 1A presents the total phenolic content (TPC), total tannin content (TTC), and total flavonoid content (TFC) of MMF. An aqueous extract of MMF had a TPC of 26.32 ± 1.05 μg GAE/mg, TTC of 28.84 ± 1.315 μg TAE/mg, and TFC of 1.92 ± 0.2 μg CE/mg. The antioxidant activity of MMF was determined by DPPH free radical scavenging assay, and their reducing ability was determined based on their concentration, providing the 50% inhibition (IC_50_) value. The maximum inhibition value was observed at an MMF concentration of 400 μg/mL. As a result, the measured inhibition concentration 50% (IC_50_) value was 213.36 ± 14.37 μg/mL (Figure 1B). In addition, an ABTS assay was performed, and the estimated IC_50_ of MMF was 833 μg/mL (Figure 1C). The superoxide anion radical scavenging activity of MMF was determined using the nitro-blue tetrazolium (NBT) reduction method. At 1 mg/mL, MMF showed a superoxide anion radical scavenging ability of approximately 30%. A slight increase was observed as the treatment concentration was increased (Figure 1D). The inhibition of the lipid peroxidation was examined using the ferric thiocyanate assay. Lipid peroxidation was prevented by 16.5% at an MMF concentration of 8 mg/mL. Lipid peroxidation has deleterious effects on cell physiology by disrupting cell signaling with reduced membrane protein activities, and MMF is believed to block this process. The Fenton reaction generates hydroxyl radicals (Figure 1E). The inhibitory effects of the MMF extract on DNA nicking are caused by hydroxyl radicals. Lanes 1, 2, 3, and 4 are pBR322 plasmid DNA, pBR322 plasmid DNA + Fenton’s reagent, pBR322 plasmid DNA + MMF (5 mg/mL) + Fenton’s reagent, and pBR322 plasmid DNA + catalase (1 unit) + Fenton’s reagent, respectively. The MMF extracts had a protective effect against hydroxyl radical-mediated plasmid DNA damage. In particular, highly reactive hydroxyl radicals ultimately cause DNA damage, confirming that MMF reduces the formation of form II (single-strand nicked DNA) and form III (double-strand nicked DNA) to form I (double-strand DNA) in the nicking of DNA mediated by Fenton’s reaction (Figure 1F). The protective effects of MMF on Cu^2+^-induced human LDL oxidation. Lanes 1, 2, 3, 4, and 5 are native LDL, LDL + Cu^2+^, LDL + Cu^2+^ + MMF (0.1 mg/mL), LDL + MMF (0.5 mg/mL) + Cu^2+^, and LDL + ascorbic acid (0.1 mg/mL) + Cu^2+^, respectively. In the aging process of blood vessels, the oxidation of low-density lipoprotein (LDL) is the most common vascular damage process known to mediate various inflammatory responses. MMF inhibited oxidized-LDL induced by Cu^2+^ in a concentration-dependent manner. The results were visualized by electrophoresis. The oxidation of apolipoprotein in LDL was demonstrated by the increased shift of bands. At 0.5 mg/mL, MMF notably protected Cu^2+^-induced LDL oxidation (Figure 1G).

### 3.2. Effects of MMF on the Viability of Murine Macrophage RAW264.7 Cells

The cytotoxic effect of MMF on the viability of RAW264.7 cells was examined using the WST assay. The RAW264.7 cells were incubated with various concentrations (0–500 μg/mL) of MMF for 24 h. No significant decrease in cell viability was observed, even at the highest concentration of MMF (Figure 2A). On the other hand, the MMF treatment dose-dependently restored the cell viability reduced by LPS (1 μg/mL) from 71.95% to 109.75% (Figure 2B). Subsequent experiments were conducted using MMF at 300 or 500 μg/mL.

### 3.3. MMF Increased the Nuclear Translocation of Nrf2 in Macrophages

The antioxidant effects of MMF on the Nrf2/HO-1 signaling pathway were investigated by western blotting. MMF dose-dependently increased HO-1 expression, and significant nuclear translocation of Nrf2 was observed at 500 μg/mL (Figure 3A). The nuclear translocation of Nrf2 was reconfirmed through immunofluorescence microscopy. MMF dose-dependently increased the nuclear translocation of Nrf2 (Figure 3B).

### 3.4. MMF Inhibited iNOS and COX-2 Expressions and the Phosphorylation of MAPKs in LPS-Stimulated Macrophages

The effects of MMF on NO production in LPS-stimulated RAW264.7 cells were evaluated. The NO levels in the medium significantly increased by LPS at 1 μg/mL for 24 h. MMF significantly and dose-dependently reduced NO production in LPS-stimulated RAW264.7 cells. At 500 μg/mL, MMF significantly decreased LPS-induced NO production from 107.66 to 17.84 μM in RAW264.7 cells (Figure 4A). The effects of MMF on the mRNA and protein expression levels of iNOS and COX2 in LPS-stimulated RAW264.7 cells were examined. The change in iNOS protein in relation to NO production was observed. MMF notably and dose-dependently inhibited the LPS-induced mRNA and protein expression levels of iNOS and COX2 in RAW264.7 cells (Figure 4B,C). The inhibitory effects of MMF on the MAPK signaling pathway in LPS-stimulated RAW264.7 cells were investigated by western blotting. LPS markedly elevated the phosphorylation level of ERK, JNK, and p38. On the other hand, high MMF concentrations significantly inhibited the phosphorylation of ERK, JNK, and p38 in LPS-stimulated RAW264.7 cells (Figure 4D).

### 3.5. MMF Inhibited the Nuclear Translocation of NFκB in LPS-Stimulated Macrophages

The effects of MMF on NFκB family members in LPS-stimulated RAW264.7 cells were investigated by analyzing the phosphorylated protein levels. Western blot images showed that MMF significantly inhibited IκB and NFκB phosphorylation by LPS (Figure 5A). Immunofluorescence microscopy showed that MMF reduced NFκB translocation to the nucleus in LPS-stimulated RAW264.7 cells compared to LPS alone (Figure 5B). Moreover, cytosolic NFκB activation was also increased. These results suggested that the MMF treatment partially affected NFκB activation in response to LPS induction through NFκB phosphorylation.

### 3.6. MMF Reduced Pro-Inflammatory Cytokine Levels in LPS-Treated Macrophages

ELISA and qPCR were performed to examine the effects of MMF on pro-inflammatory cytokine production. LPS increased TNFα, IL-1β, and IL-6 production in RAW264.7 cells. ELISA showed that LPS stimulation significantly upregulated the production of the pro-inflammatory cytokines (TNFα, IL-1β, and IL-6) in RAW264.7 cells. On the other hand, at 500 μg/mL, MMF significantly downregulated the TNFα, IL-1β, and IL-6 levels induced by LPS (Figure 6A). The qPCR results also showed that MMF dose-dependently decreased the mRNA expression levels of TNFα, IL-1β, and IL-6 (Figure 6B). These data suggest that MMF exerts anti-inflammatory activity by suppressing pro-inflammatory cytokines in LPS-stimulated RAW264.7 cells.

### 3.7. MMF Decreased NO Production in LPS-Stimulated Zebrafish

The inhibitory effects of MMF on NO production in LPS-induced zebrafish larvae were evaluated using the fluorescent probe DAF-FM DA. The LPS treatment induced NO production in zebrafish larvae, and MMF reduced this LPS-induced increase in NO production (Figure 7).

### 3.8. MMF Improved Body Weight and Serum Lipid Levels

The zebrafish study evaluated the benefits of MMF on high-cholesterol diet-induced obesity in adult zebrafish. There was no difference between the HCD-fed groups (HCD, HCD + MMF 10%, and HCD + MMF 20%) until the first week, but a lower BMI was observed in the MMF intake group after the second week. On the sacrifice day, a difference was observed between mean body weights in the three groups (NCD, HCD + MMF 10%, and HCD + MMF 20%) and the HCD group (Figure 8A). The plasma total cholesterol levels in adult zebrafish fed on the HCD showed similar metabolic features to human obesity. MMF significantly and dose-dependently improved the plasma total cholesterol in high-cholesterol diet-induced obesity in adult zebrafish (Figure 8B).

### 3.9. HPLC Analysis Indicates the Potential Presence of Quercetin

The HPLC analysis was performed with MMF extract and previously known ingredients of MMF. HPLC chromatogram showed a peak of an identified standard compound of quercetin at a retention time of 2.711 min (Figure 9A). This peak corresponded to the peak in MMF used in this study at a retention time of 2.747 min (Figure 9B). We estimated the concentration of quercetin contents in MMF extract as 891.9 μg/g (*w*/*w*) based on the standard curve quantification.

## 4. Discussion

Metabolic disorders are complex conditions characterized by abnormal metabolic profiles and are closely related to diabetes, obesity, and hyperlipidemia [46]. Hyperglycemia, a hyperlipidemia condition of the host, directly damages the systemic oxidant/antioxidant balance in vascular epithelial tissue or immune cells and aggravates the inflammatory status [47]. A closer look showed that a low-grade inflammatory status in adipose tissue stimulates preadipocyte hyperplasia and hypertrophy [48]. On the other hand, excessive ROS generated from beta-oxidation mechanisms of adipocytes underlies the cellular pathophysiology of these metabolic diseases. ROS generation activates multifarious inflammatory signaling pathways and promotes inflammatory response [49]. Inflammation and oxidative stress play a significant role in the development of metabolic comorbidities such as hyperlipidemia, hypertension, and glucose intolerance, which lead to metabolic dysfunction [50,51]. The trio of oxidative stress, inflammation, and adiposity make it tricky to stop the vicious cycle of metabolic disorders. A functional food or medicine with this multifaceted potential might efficiently impact patients with metabolic disorders. Health-functional foods derived from natural resources have multifaceted potentials, such as nutritional value and pharmacological properties [52].

In this study, the potential of MMF as a functional food was evaluated using a series of experiments, including chemical assays and in vitro and in vivo system-based studies. MMF was highly expected to have antioxidant ability by evaluating the phenol, tannin, and flavonoid contents. The free radical scavenging ability was evaluated. The IC_50_ value obtained in the DPPH assay revealed no significant effect in the nitroblue tetrazolium (NBT) assay. Therefore, MMF is believed to scavenge cation radicals rather than anion radicals. Reactive oxygen species are a major source of oxidative and nitrosative stress for organisms, damaging macromolecules, including lipids and DNA [53]. The antioxidant activity of MMF during linoleic acid oxidation, as measured by the thiocyanate method, did not increase by more than 16.5%. On the other hand, the antioxidant activity of MMF during LDL oxidation was significant, suggesting that the antioxidant activity of natural antioxidants is system-dependent and that a wide range of activities can be observed according to the lipid systems used as substrates. This study evaluated the potential of MMF as an antioxidant defense against hydroxyl radical-mediated plasmid DNA damage using agarose gel electrophoresis. The results show that MMF water extract protects against DNA damage induced by metal-catalyzed Fenton reactions. The antioxidants protect DNA from oxidative damage, changing the three-dimensional structure of DNA and altering DNA mobility in electric fields. The plasmid DNA appears in three distinct forms on an agarose gel. Form I is a supercoiled circular form that migrates faster than the other forms. A nicked circular form (form II) is formed when the supercoiled DNA form is broken. This form moves much more slowly than the other forms. Another form is Form III, a linear form between Forms I and II. MMF significantly minimized this cleavage.

The inflammatory response is a complex biological mechanism of the host’s immune system that defends the body against harmful stimuli. On the other hand, persistent inflammation can lead to diseases and tumorigenesis, such as atherosclerosis, rheumatoid arthritis, and cancer [54]. During an inflammatory immune response, macrophages release pro-inflammatory mediators, such as NO, iNOS, COX2, and pro-inflammatory cytokines and chemokines. MMF reduced iNOS and COX-2 expression and the release of cytokines, such as TNFα, IL-1β, and IL-6, in LPS-stimulated macrophages [55,56]. Inhibiting the production of inflammatory mediators might be a useful therapeutic strategy in inflammatory diseases. NFκB regulates the transcription of inflammatory factors, and its activation is regulated by a MAPK (mitogen-activated protein kinase) and is involved in the inflammatory process through nuclear translocation [57,58]. In this study, using RAW264.7 cells, MMF attenuated LPS-induced phosphorylation of the NFκB pathway proteins and the nuclear translocation of p65. The MAPK pathway regulates various metabolic processes of cells and is strongly involved in the immune defense and inflammatory response. MMF significantly decreased the phosphorylation of LPS-stimulated MAPK pathway protein expression.

This study applied an inflammation model using LPS and an obesity model using an HCD diet in zebrafish. The zebrafish model is used widely in studies to screen drugs with anti-inflammatory properties because it acts as an in vivo inflammation model exhibiting similar inflammatory responses to mammals [59]. In addition, zebrafish are increasingly utilized as models of human obesity and metabolic diseases, including visceral adiposity, hepatic steatosis, atherosclerosis, and type 2 diabetes [60]. MMF inhibited LPS-induced NO production in vivo embryos, as shown by DAF-FM DA fluorescence analysis. Moreover, the BMI and TC of zebrafish were reduced dose-dependently by the MMF treatment. In addition to the prominent anti-inflammatory properties in vitro, the MMF was assumed to exert weight reduction and hypocholesterolemic effects by controlling the systemic oxidative and inflammatory status.

As fermentation progresses, crude phytochemical components are metabolized into bioactive substances by bacteria and fungi, which benefit human health. For example, MMF is reported to be habituated by numerous resident bacteria, and the fermentation progressed by dominant bacteria genera of Enterobacter, Pediococcus, Pseudomonas, Mucor, and Saccharomyces [61]. Lactic acid was produced by fermentation [30], and many studies reported the benefits of lactic acid or lactic acid bacteria-derived metabolites against obesity or metabolic diseases [62].

Quercetin, which is universally found in numerous herbs, can be produced from its various glycoside forms during the fermentation process [29]. Several studies have shown that MMF contains quercetin as a major compound [63]. This bioactive aglycone of quercetin has been reported to exert potent antioxidant activity, thereby providing multifaceted benefits for metabolic disorders [64]. As presented in the HPLC analysis, our MMF extract contains quercetin, which is equivalent to around 1.49 μM, which may be contributing to the mechanisms of systemic effects.

The standardization of MMF is indeed a significant issue for researchers. Consequently, many studies have investigated the major compounds of MMF, particularly following the fermentation process. Major composition profiles of MMFs from different batches have been investigated by researchers. A recent study investigated the differences among nine MMFs produced from various batches in China and Korea [65]. While it is not a simple issue, we may assert that many studies have been conducted and are currently underway by researchers to address the standardization issues of MMF.

This study has several limitations in revealing the detailed mechanisms for the anti-obesity effects, e.g., appetite suppression and enhancement of metabolic rate or inhibition of nutrient digestion, because of the technical limitation of the zebrafish model. On the other hand, MMF may exert systemic benefits on HCD-fed zebrafish, as shown by fluorescence imaging. Nevertheless, no zebrafish in the MMF group (in 10% and 20%) expired, indicating the low toxic profile of MMF in vivo. A low toxicity profile is a critical factor for a new food material to be developed as a functional food [66]. We can reasonably assume that at least quercetin, as detected in our MMF sample, may help mitigate HCD or LPS-induced harmful conditions in zebrafish due to its notable antioxidant activities.

MMF, a fermented food used in traditional medicine, was evaluated as a potential modern health functional food. It showed notable potential for metabolic disorders and reliable background effects on treating oxidative and inflammatory conditions. Thus, MMF may have high value as a healthy, functional food, but additional research is needed.

## 5. Conclusions

MMF can scavenge free radicals, activate the Nrf2/HO-1 pathway in RAW264.7 cells, and show anti-inflammatory effects by attenuating the production of the pro-inflammatory mediators by inhibiting NFκB and MAPK signaling activation. In a metabolic disorder model using zebrafish, the BMI and TC of MMF groups were reduced significantly compared to the HCD group. These results highlight the potential of MMF as a healthy, functional food.

## Figures and Tables

**Figure 1 antioxidants-13-01271-f001:**
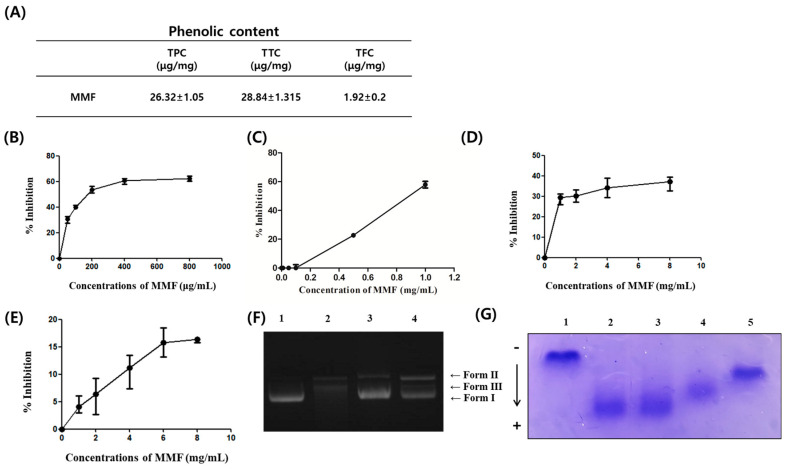
Phenolic contents and antioxidant activity of the MMF extract. (**A**) Total phenolic, tannin, and flavonoid contents of MMF extract. (**B**,**C**) Free-radical scavenging activity of MMF extract measured using a DPPH assay and ABTS assay. (**D**) The inhibitory effect of MMF extract was tested by NBT reduction system. (**E**) Lipid peroxidation inhibitory effect of MMF. (**F**) Showing a comparative electrophoretic pattern of pBR322 DNA nicking inhibition activity of MMF. Lane 1, pBR322 supercoil plasmid DNA; Lane 2, hydroxyl radical-mediated DNA nick form; Lane 3, pre-treatment of MMF (5 mg/mL, final concentration); Lane 4, pre-treatment of catalase (1 unit). (**G**) Effect of MMF on the Cu^2+^ mediated LDL oxidation by relative electrophoretic mobility (REM). Lane 1, native LDL; Lane 2, ox-LDL; Lane 3, pre-treatment of MMF (0.1 mg/mL); Lane 4, pre-treatment of MMF (0.5 mg/mL); Lane 5, pre-treatment of ascorbic acid (0.1 mg/mL). Dose-dependent effect of MMF against Cu^2+^-mediated LDL oxidation by REM.

**Figure 2 antioxidants-13-01271-f002:**
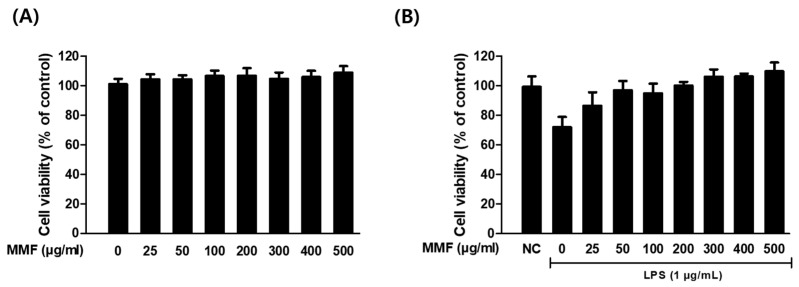
Effects of MMF on cell viability. The cells were treated with various concentrations of MMF extract in the absence (**A**) or presence (**B**) of LPS (1 μg/mL) for 24 h. The results are presented as the mean ± SDs of the percentages determined by three independent experiments versus non-treated controls. NC: non-treated negative control.

**Figure 3 antioxidants-13-01271-f003:**
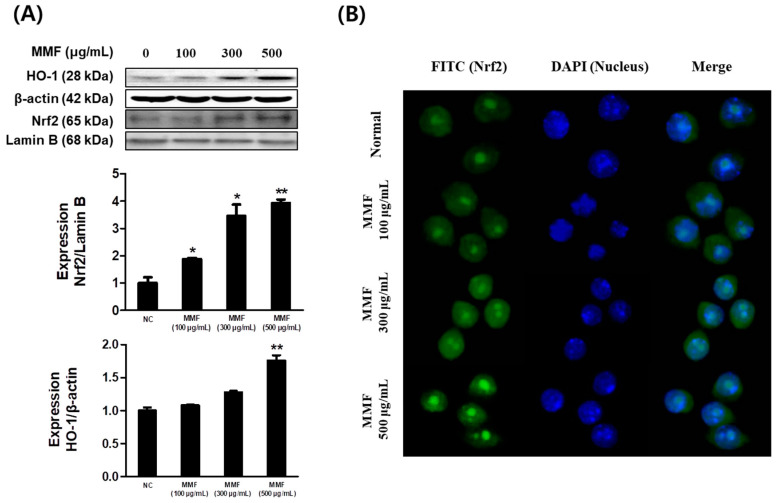
Effects of MMF on HO-1 expression and the nuclear translocation of Nrf2. (**A**) Raw 264.7 cells were incubated with various concentrations of MMF for 6 h and HO-1 expression was examined by western blot analysis. The Nrf2 expression levels in cytoplasm and nuclear were assessed by western blot analysis. (**B**) Nuclear Nrf2 translocation was visualized under an immunofluorescence microscope. The band intensities were measured by densitometry and normalized to the intensities of the total forms and β-actin. The results are presented as the means ± SDs of three independent experiments. * *p* < 0.05, ** *p* < 0.01 versus non-treated negative control. NC: non-treated negative control.

**Figure 4 antioxidants-13-01271-f004:**
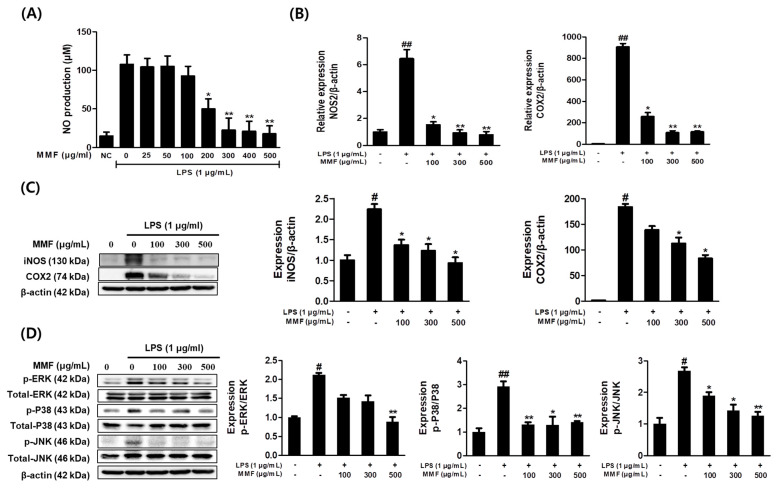
MMF inhibited the up-regulation of iNOS and COX2 and the phosphorylation of MAPKs in LPS-treated RAW264.7 macrophages. RAW264.7 macrophages were co-treated with MMF and LPS (1 μg/mL) for 12–24 h. (**A**) NO concentrations were estimated using a Griess reaction. (**B**) Gene expression levels of NOS2 and COX2 obtained by Real-time PCR (**C**) Western blot analysis revealed the effects on the relative iNOS and COX2 protein levels. (**D**) Relative MAPK expression determined by western blot analysis. The band intensities were measured by densitometric analysis and normalized versus the intensities of the total forms and β-actin. The results are presented as the means ± SDs of three independent experiments. # *p* < 0.05, ## *p* < 0.01 versus LPS-treated controls, and * *p* < 0.05, ** *p* < 0.01 versus LPS-treated RAW264.7 cells. NC: non-treated negative control.

**Figure 5 antioxidants-13-01271-f005:**
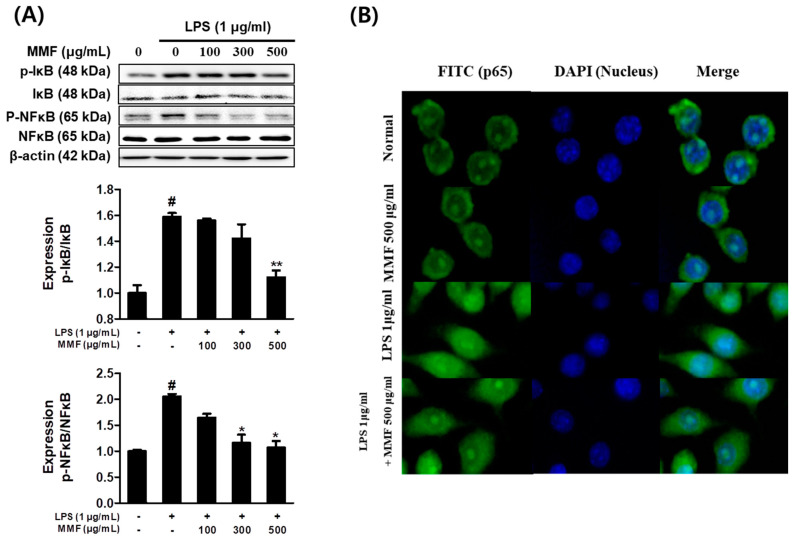
MMF inhibited NFκB translocation by reducing the phosphorylation of IκB in RAW264.7 macrophages. RAW264.7 macrophages were co-treated with MMF and LPS (1 μg/mL) for 12 h. (**A**) Relative phosphorylation of the IκB and NFκB protein levels as determined by western blot analysis. (**B**) visualized by immunofluorescence microscopy. The results are presented as the means ± SDs of three independent experiments. # *p* < 0.05 versus LPS-treated controls, and * *p* < 0.05, ** *p* < 0.01 versus LPS-treated RAW264.7 cells.

**Figure 6 antioxidants-13-01271-f006:**
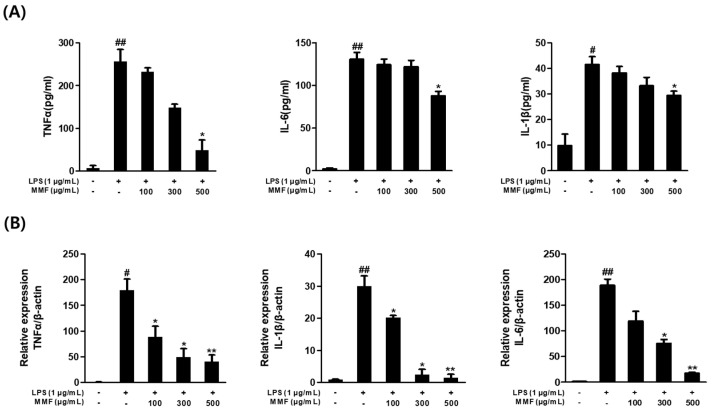
MMF significantly reduced LPS-induced pro-inflammatory cytokine levels in RAW264.7 macrophages. RAW264.7 macrophages were simultaneously treated with MMF and LPS (1 μg/mL) for 24 h. (**A**) Relative expression of TNFα, IL-1β, and IL-6, as determined by ELISA. (**B**) Relative expression of TNFα, IL-1β, and IL-6, as determined by qPCR. The results are presented as the means ± SDs of three independent experiments. # *p* < 0.05; ## *p* < 0.01 versus LPS-treated controls, and * *p* < 0.05; ** *p* < 0.01 versus LPS-treated RAW264.7 cells.

**Figure 7 antioxidants-13-01271-f007:**
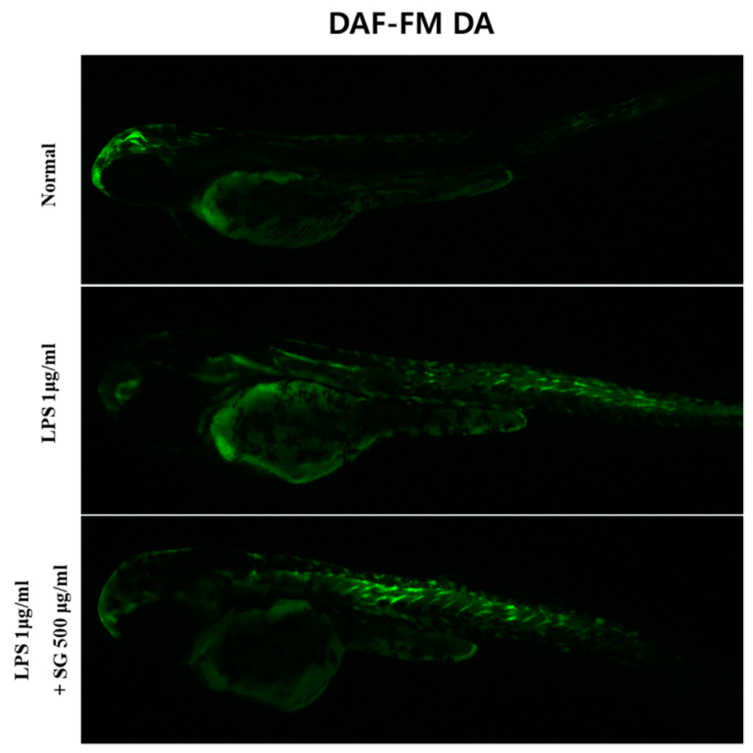
Protective effects of MMF on LPS-induced NO generation in zebrafish larvae. The zebrafish larvae were pre-treated with MMF (500 μg/mL) for 1 h and then exposed to LPS (10 μg/mL) for 24 h. The levels of NO generation were observed under a fluorescence microscope after staining with DAF-FM DA.

**Figure 8 antioxidants-13-01271-f008:**
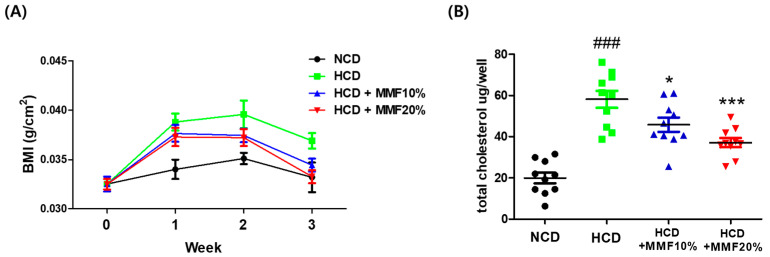
Assessment of the BMI and plasma total cholesterol in adult zebrafish fed MMF. (**A**) Change in BMI (g/cm^2^) during the experimental period for three weeks. (**B**) TC levels in HCD-induced obese zebrafish at three weeks fed 10% and 20% MMF + HCD. NCD: normal cholesterol diet; HCD: high-cholesterol diet. * *p* < 0.05, *** *p* < 0.001, ### *p* < 0.001.

**Figure 9 antioxidants-13-01271-f009:**
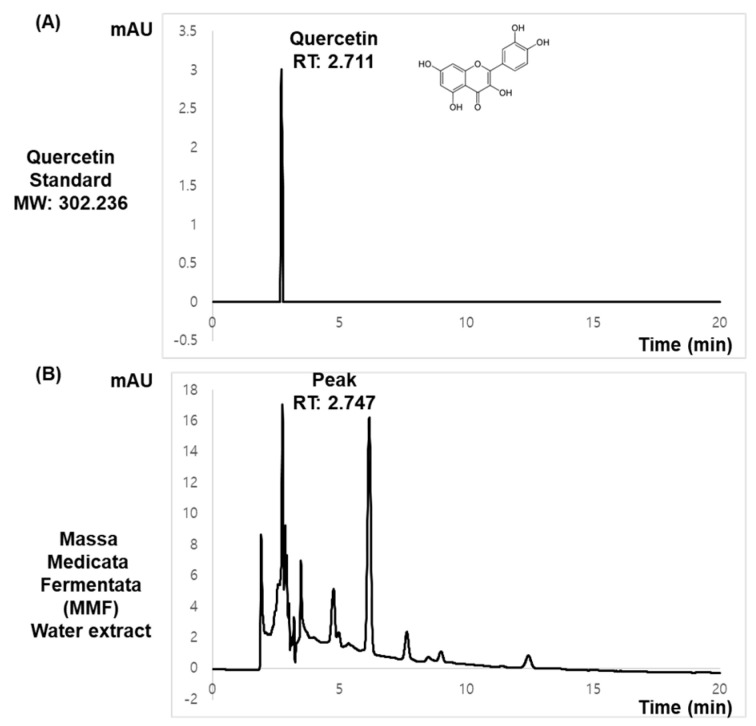
HPLC fingerprinting analysis of MMF extract. (**A**) HPLC chromatogram analysis of Quercetin, a standard compound of MMF. (**B**) HPLC analysis of hydrothermal extract of MMF used in this study.

## Data Availability

Data are contained within the article.

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
