# Peer review of "Massa Medicata Fermentata, a Functional Food for Improving the Metabolic Profile via Prominent Anti-Oxidative and Anti-Inflammatory Effects"

_antioxidants, 2024, doi:10.3390/antiox13101271_

Round 1
Reviewer 1 Report
This paper investigates the antioxidant and anti-inflammatory properties of MMF through both in vitro and in vivo experiments. Overall, the study is well-designed and well-structured, and the findings contribute valuable insights toward understanding the health benefits of MMF. However, I have the following suggestions that may improve the quality and impact of the paper:
1. While the paper effectively demonstrates the potential of MMF as a functional food, it lacks a clear discussion on how these findings could be translated into human health applications. Are there any existing studies on MMF or similar compounds in humans? Addressing this would strengthen the relevance of the study for clinical and practical use.
2. The mechanistic details of the study are somewhat superficial. For example, the role of quercetin is mentioned, but its interactions with specific molecular targets are not fully elucidated. Further exploration of these pathways would provide a more comprehensive understanding of MMF's bioactivity.
3. Given the emphasis on quercetin’s significance in the study, it is necessary to provide the absolute amount of quercetin present in MMF. This will allow for a clearer assessment of its contribution to the overall effects observed.
4. The statement that "MMF is believed to scavenge cation radicals rather than anion radicals" (line 533) based solely on the DPPH and NBT assay results appears premature. Phenolic compounds, including those in MMF, can scavenge both cationic and anionic radicals, though their efficacy may vary depending on the type of radical and the assay used. It would be more accurate to suggest that MMF may exhibit stronger activity against certain types of radicals, but further studies using additional radical scavenging assays (e.g., ABTS, hydroxyl radicals) are required to draw a more definitive conclusion.
Author Response
- While the paper effectively demonstrates the potential of MMF as a functional food, it lacks a clear discussion on how these findings could be translated into human health applications. Are there any existing studies on MMF or similar compounds in humans? Addressing this would strengthen the relevance of the study for clinical and practical use.
- We are greatly honored to contribute to the special issue of the renowned journal of Antioxidants. We hope you find the revised version satisfactory.
- Several articles introduce MMF as a food or functional food [1] [2] [3], emphasizing the potential benefits of the fermentation process of MMF.
- Although the efficacy of MMF has long been recognized by Traditional Chinese Medicine (TCM) practitioners, the preparation methods often vary across the literature, leading to challenges in standardizing its production and commercialization [1].
- Additionally, its medicinal properties have not been extensively studied in clinical trials. No studies have been reported on the clinical efficacy of MMF alone in human health..
- However, we found some clinical studies that used prescriptions containing MMF.
- Ha et al reported the effect of Naesohwajung-tang on functional dyspepsia in randomized, double blind, placebo-controlled, multi-center trial [4].
- Kim et al reported seven clinical NERD (non-erosive gastro-esophageal reflux disease) cases treated with Ljintang-Gamibang, an herbal prescription which also contains MMF as major ingredient[5].
- Despite these studies, research on the mechanisms and the efficacy of MMF in metabolic diseases remains limited. Therefore, we believe the findings of our study offer novel insights.
- The mechanistic details of the study are somewhat superficial. For example, the role of quercetin is mentioned, but its interactions with specific molecular targets are not fully elucidated. Further exploration of these pathways would provide a more comprehensive understanding of MMF's bioactivity.
- We sincerely appreciate your thoughtful comments.
- Quercetin is a well-known and widely distributed compound found in numerous plants. Many studies have elucidated the detailed mechanisms of quercetin through extensive research by numerous scientists. As an evidence for that, searching for papers with keyword “quercetin LPS raw 264.7 antioxidant pathway” in Google scholar shows the 14,400 searches.
- For example, Lee et al. demonstrated that the anti-inflammatory effects of quercetin in LPS-treated RAW 264.7 cells are mediated through the molecular targets NF-kB, ERK1/2, JNK, and others [6].
- Additionally, Byun et al. reported that quercetin negatively regulates the toll-like receptor 4 (TLR4) pathway [7].
- A recent study by Xu et al. also found that quercetin prevents LPS-induced oxidative stress by interacting with the AKT-FOXO1 and KEAP1-NRF2 signaling pathways [8].
- We would like to encourage the reviewer to focus on the reported benefits and effects of MMF, rather than quercetin, as extensive studies on quercetin already exist.
- Given the emphasis on quercetin’s significance in the study, it is necessary to provide the absolute amount of quercetin present in MMF. This will allow for a clearer assessment of its contribution to the overall effects observed.
- We agree with your suggestion that quantifying the quercetin content in MMF is necessary.
- Therefore, we conducted an additional study to generate a standard curve for quercetin at various concentrations. We estimated the quercetin content in our MMF samples (dried extract) to be approximately 891.9 μg/g (w/w).
- The quercetin content has been incorporated into the revised Figure 9.
- The statement that "MMF is believed to scavenge cation radicals rather than anion radicals" (line 533) based solely on the DPPH and NBT assay results appears premature. Phenolic compounds, including those in MMF, can scavenge both cationic and anionic radicals, though their efficacy may vary depending on the type of radical and the assay used. It would be more accurate to suggest that MMF may exhibit stronger activity against certain types of radicals, but further studies using additional radical scavenging assays (e.g., ABTS, hydroxyl radicals) are required to draw a more definitive conclusion.
- We appreciate your valuable suggestion to include the ABTS assay for evaluating the antioxidant properties of MMF.
- In response to your request, we conducted the ABTS assay. The estimated IC50 of radical scavenging activity is 833 μg/ml, which is higher than the IC50 observed in the DPPH scavenging assay.
- We have updated Figure 1 to include the new data from the ABTS assay (Figure 1C). Additionally, we have revised the figure legend, methods, and results sections to incorporate the new findings from the ABTS assay.
Remarks
- We read the manuscript thoroughly and corrected sentences as per the editor’s request.
We really hope the reviewers satisfied with our revised manuscript. Thank you very much.
References
- Xu, M.-S.; Fu, Q.; Baxter, A. The components and amylase activity of Massa Medicata Fermentata during the process of fermentation. Trends in Food Science & Technology 2019, 91, 653-661.
- Fu, F.Q.; Xu, M.; Wei, Z.; Li, W. Biostudy on traditional Chinese medicine massa medicata fermentata. ACS omega 2020, 5, 10987-10994.
- Bai, Y.; Zheng, M.; Fu, R.; Du, J.; Wang, J.; Zhang, M.; Fan, Y.; Huang, X.; Li, Z. Effect of Massa Medicata Fermentata on the intestinal flora of rats with functional dyspepsia. Microbial Pathogenesis 2023, 174, 105927.
- Ha, N.-Y.; Ko, S.-J.; Park, J.-W.; Kim, J. Efficacy and safety of the herbal formula Naesohwajung-tang for functional dyspepsia: a randomized, double-blind, placebo-controlled, multi-center trial. Frontiers in Pharmacology 2023, 14, 1157535.
- Kim, B.-s.; Lim, H.-y.; Oh, J.-h.; Kim, D.-w.; Choi, B.-h.; Hur, J.-i.; Kim, D.-j.; Byun, J.-s. Seven cases of non-erosive gastroesophageal reflux disease who were treated by Ljintang-Gamibang and acupuncture. The Journal of Internal Korean Medicine 2005, 26, 926-934.
- Lee, H.N.; Shin, S.A.; Choo, G.S.; Kim, H.J.; Park, Y.S.; Kim, B.S.; Kim, S.K.; Cho, S.D.; Nam, J.S.; Choi, C.S. Anti‑inflammatory effect of quercetin and galangin in LPS‑stimulated RAW264. 7 macrophages and DNCB‑induced atopic dermatitis animal models. International Journal of Molecular Medicine 2018, 41, 888-898.
- Byun, E.-B.; Yang, M.-S.; Choi, H.-G.; Sung, N.-Y.; Song, D.-S.; Sin, S.-J.; Byun, E.-H. Quercetin negatively regulates TLR4 signaling induced by lipopolysaccharide through Tollip expression. Biochemical and biophysical research communications 2013, 431, 698-705.
- Xu, J.; Li, Y.; Yang, X.; Li, H.; Xiao, X.; You, J.; Li, H.; Zheng, L.; Yi, C.; Li, Z. Quercetin inhibited LPS-induced cytokine storm by interacting with the AKT1-FoxO1 and Keap1-Nrf2 signaling pathway in macrophages. Scientific Reports 2024, 14, 20913.

Reviewer 2 Report
I would like to emphasize the impressive amount of work, and the results of high quality presented in this paper. This study was carried out under very good experimental conditions and is presented in a clear and precise manner. It therefore deserves to be published in this journal.
Nevertheless, I have two important criticisms, and the authors should address these questions and provide answers in their paper.
Being unaware of this product until reading this article, I wonder about its molecular composition, stability over time and batch to batch variation. The authors should address these important items. The fact that MMF originates from a mixture of natural products – therefore, with usual variations – and is fermented by microorganisms strongly suggests that its composition should vary from batch to batch. It is therefore highly probable that the results that ae presented in this paper are representative only of this batch and could not be generalized to all MMF on the market. I, however, easily consider that, in an academic context, the authors focused on a complete study of one batch and did not compare several batches on a limited number of parameters. Nevertheless, with a potential application for human health, comparing a few batches on a limited number of parameters – f.i. content of quercetin, NO production in RAW264.7 cells, … – should strengthen the scope of the work.
The doses of MMF required to gain the benefic effects is puzzling. It is a recurrent drawback with studies using cultured cells – excepted for those originating from the gastro-intestinal barrier – that the bioaccessibility and the bioavailability are not taken into account. Clearly, no data are provided on the nature and the amount of the active molecules present in the MMF that will really gain access to the macrophages, RAW264.7 cells in this model, and this could considerably affect the results. Furthermore, the concentrations that were required to obtain some of the effects would probably never be reached in plasma. Nevertheless, the data obtained in Zebrafish larvae suggest effects in vivo, although this interesting model is not fully representative of human beings.
Two minor remarks.
1. Fig. 2: NC should be defined in the legend.
2. About NFkB and iKB, in the title of 3.5 section and the legend of Figure 5, “suppressed” should be replaced by “inhibited”.
Author Response
I would like to emphasize the impressive amount of work, and the results of high quality presented in this paper. This study was carried out under very good experimental conditions and is presented in a clear and precise manner. It therefore deserves to be published in this journal.
Nevertheless, I have two important criticisms, and the authors should address these questions and provide answers in their paper.
- First of all, it is a great honor to have our manuscript reviewed by your esteemed journal. We hope that, with the guidance of your excellent editors and reviewers, the quality of our research will meet the high standards required for publication in this journal.
1, Being unaware of this product until reading this article, I wonder about its molecular composition, stability over time and batch to batch variation. The authors should address these important items. The fact that MMF originates from a mixture of natural products – therefore, with usual variations – and is fermented by microorganisms strongly suggests that its composition should vary from batch to batch. It is therefore highly probable that the results that ae presented in this paper are representative only of this batch and could not be generalized to all MMF on the market.
- We are grateful for your thoughtful and important comments, which highlight the challenges in the standardization of MMF as a herbal medicine.
- As you pointed out, the standardization of MMF is indeed a significant issue for researchers. Consequently, there are many studies that investigated the major compounds of MMF, particularly following the fermentation process.
- In Chinese science literature archive (Chinese National Knowledge Infrastructure, CNKI. Written in Chinese) alone, 45 papers can be found with search query “Massa Fermentata”. Many of these are original articles that exploring the molecular composition of MMF, with several comparing its composition before, during, and after fermentation. Most of these papers are not searched in Google Scholar.
- We have arranged the search results from CNKI as table below for reviewer. These references are listed separately in the last part of this review report.
Title |
Author / Journal/ Year |
Table Reference |
Rapid Analysis of Chemical Components in Massa Medicata Fermentata Based on UHPLC-LTQ-Orbitrap-MS/MS Technology |
Du et al./ Chinese Journal of Modern Applied Pharmacy/ 2024 |
[1] |
Determination of 4 Alkylresorcinols in Massa Medicata Fermentata by Solid Phase Extraction-Ultra Performance Liquid Chromatography-Tandem Mass Spectrometry |
Li et al./ Analysis and Testing Technology and Instruments/2024 |
[2] |
Correlation between changes in enzymatic properties and microbial communities during fermentation of Massa Medicata Fermentata |
Ding et al./ Chinese Journal of Hospital Pharmacy/2024 |
[3] |
Effects of different fermentation conditions on content changes of five components in Massa Medicata Fermentata produced in Zhejiang Province |
Zhang et al./China Medical Herald/2024 |
[4] |
Rapid Identification and Bioactivity Study of Chemical Constituents from the Massa Medicata Fermentata |
Zhang et al./ Chinese Journal of Modern Applied Pharmacy/2024 |
[5] |
Variations of glucose content in Massa Medicata Fermentata during processing based on quantitative proton nuclear magnetic resonance |
Shi et al/ China Journal of Chinese Materia Medica/2023 |
[6] |
Analysis of volatile matter changes of Massa Medicata Fermentata before and after processing based on HS-GC-IMS technology |
Shi et al/ Chinese Traditional and Herbal Drugs/2023 |
[7] |
Chemical constituents from Massa Medicata Fermentata and their antioxidation activities |
Li et al/ Chinese Traditional Patent Medicine/2023 |
[8] |
Study on the chemical constituents of Massa Medicata Fermentata before and after fermentation based on UHPLC-Q-Orbitrap |
Gao et al./ Chinese Traditional Patent Medicine/2022 |
[9] |
- While it is not a simple issue, we may assert that many studies have been conducted and are currently underway by researchers to address the standardization issues of MMF. I am optimistic that the standardization issue of MMF will be resolved in the near future.
I, however, easily consider that, in an academic context, the authors focused on a complete study of one batch and did not compare several batches on a limited number of parameters. Nevertheless, with a potential application for human health, comparing a few batches on a limited number of parameters – f.i. content of quercetin, NO production in RAW264.7 cells, … – should strengthen the scope of the work.
- We greatly appreciate your time and effort in reviewing our manuscript.
- We completely agree that the varying compositions of MMF from different batches, resulting from unstandardized fermentation processes, must be considered in studies, as this variability can influence the efficacy of MMF and impact the results.
- However, in the context of our study, it would be questionable to derive meaningful conclusions from comparing the effects of MMFs from different batches. Since our focus is not on the fermentation/processing process, the study design is not suited for a comparative analysis, and such an approach could undermine the coherence of the manuscript.
- First, preparing those batches into samples within the provided revision period of 10 days is not possible, as several steps are required to purchase different batches and extract compounds from raw herbal materials.
- Second, every herbal medicine inherently has batch variability issues. Many herbal medicines undergo various processing steps after the raw materials are harvested, including fermentation, drying, steaming, frying, and even charring. Each of these processes can contribute to batch differences. If we were to request batch comparison studies for all research involving herbal medicines, it would lead to significant inefficiencies.
- Third, there are already several papers that address similar questions regarding MMF batch comparisons. Therefore, we believe it is reasonable to cite relevant references that cover this scope and explain the outcomes of those studies.
- A recent study investigated the differences among nine MMFs produced from various batches in China and Korea [9]. This study compared microbial constituents, organic acid contents, enzymatic activities, and 39 volatile compounds, revealing both commonalities and diversity among MMF products.
- Meanwhile, another study examined the differences between MMF samples obtained at various fermentation times [1]. This study analyzed the contents of laetrile, benzaldehyde, and rutin, as well as enzymatic activities, concluding that the chemical constituents of MMF were influenced by microorganisms.
- Additionally, a study assessed the enzymatic activities of MMF prepared with different proportions of wheat flour and varying fermentation times [10]. Using the optimal wheat flour mix ratio, the authors tracked changes in enzymatic activity over the fermentation period.
- Another study conducted in 2022 investigated the fermentation characteristics and chemical components resulting from the fermentation process in MMF. They prepared and tested six batches of MMF during this study [11].
- We respectfully request that the reviewer consider these previous references that address the batch variability issues of MMF.
2, The doses of MMF required to gain the benefic effects is puzzling. It is a recurrent drawback with studies using cultured cells – excepted for those originating from the gastro-intestinal barrier – that the bioaccessibility and the bioavailability are not taken into account.
Clearly, no data are provided on the nature and the amount of the active molecules present in the MMF that will really gain access to the macrophages, RAW264.7 cells in this model, and this could considerably affect the results.
Furthermore, the concentrations that were required to obtain some of the effects would probably never be reached in plasma.
Nevertheless, the data obtained in Zebrafish larvae suggest effects in vivo, although this interesting model is not fully representative of human beings.
- We are pleased that the reviewer highlighted essential and fundamental points regarding the gaps between in vitro and in vivo systems.
- We would like to emphasize that our findings, like those of other in vitro and in vivo studies conducted by fellow researchers, do not guarantee definitive efficacy in humans. However, pre-clinical studies (both in vitro and in vivo) play a crucial role in screening more potent materials for advancement to clinical studies.
- As researchers, we fully understand that there are significant and unavoidable differences between the two systems (in vitro and in vivo). As you pointed out, a major gap exists because in vitro experimental systems do not take into account ADME (absorption, distribution, metabolism, and excretion).
- However, we can also estimate the quantity of quercetin present in culture media following MMF treatment. We have reinforced the HPLC data to present the estimated concentration of quercetin in our MMF sample as 891.9 μg/g (w/w) (revised Figure 9). This is equivalent to approximately 0.9 μg of quercetin per well (2 mL), which translates to around 1.49 μM. For reference, a study has reported that the IC50 of quercetin is estimated to be around 1.84 μg/ml, as determined by the DPPH assay (antioxidant effect) [12].
- While we attribute our findings of MMF to the effect of quercetin, it is important to note that MMF contains a variety of physiologically active substances. Therefore, we believe that the actual antioxidant effect of MMF is likely to be greater than what we have estimated only with quercetin.
- Pharmacokinetic studies are generally not performed using zebrafish models. Therefore, we cannot assure that the treated MMF is actually absorbed and circulating in the plasma of zebrafish.
- Our search revealed no studies investigating the pharmacokinetic properties of MMF in vivo or in clinical trials. However, several ADME studies provide some insights into these issues. For example, human pharmacokinetics study [13], rat pharmacokinetics study [14], mice absorption and disposition study [15] have been conducted with quercetin, a bioactive compound found in MMF, which has been detected in plasma levels following oral administration..
- Furthermore, there are numerous reports indicating that dietary quercetin exerts beneficial effects on zebrafish under various conditions, such as hepatic oxidative damage [16], immune response and inflammation [17], even on alzheimer’s disease [18]. Therefore, we can reasonably assume that at least quercetin, as detected in our MMF sample, may help mitigate HCD or LPS-induced harmful conditions due to its notable antioxidant activities.
- We hope reviewer is satisfied with our explanations. Thank you very much.
Two minor remarks.
- Fig. 2: NC should be defined in the legend.
- We have corrected the figure legends throughout the manuscript.
- Additionally, we have added the following sentences in the relevant legends:
“NCD: normal cholesterol diet, HCD: high-cholesterol diet” in Figure 8..
- About NFkB and iKB, in the title of 3.5 section and the legend of Figure 5, “suppressed” should be replaced by “inhibited”.
- Thank you for your suggestions regarding word corrections in our manuscript; we have addressed them promptly.
We are confident that our work will contribute to a better understanding of the efficacy and potential of MMF as a functional food with antioxidant effects, supported by appropriate evidence.
Remarks
- We read the manuscript thoroughly and corrected sentences as per the editor’s request.
- We really hope the reviewers satisfied with our revised manuscript. Thank you very much.
References
- Xu, M.-S.; Fu, Q.; Baxter, A. The components and amylase activity of Massa Medicata Fermentata during the process of fermentation. Trends in Food Science & Technology 2019, 91, 653-661.
- Fu, F.Q.; Xu, M.; Wei, Z.; Li, W. Biostudy on traditional Chinese medicine massa medicata fermentata. ACS omega 2020, 5, 10987-10994.
- Bai, Y.; Zheng, M.; Fu, R.; Du, J.; Wang, J.; Zhang, M.; Fan, Y.; Huang, X.; Li, Z. Effect of Massa Medicata Fermentata on the intestinal flora of rats with functional dyspepsia. Microbial Pathogenesis 2023, 174, 105927.
- Ha, N.-Y.; Ko, S.-J.; Park, J.-W.; Kim, J. Efficacy and safety of the herbal formula Naesohwajung-tang for functional dyspepsia: a randomized, double-blind, placebo-controlled, multi-center trial. Frontiers in Pharmacology 2023, 14, 1157535.
- Kim, B.-s.; Lim, H.-y.; Oh, J.-h.; Kim, D.-w.; Choi, B.-h.; Hur, J.-i.; Kim, D.-j.; Byun, J.-s. Seven cases of non-erosive gastroesophageal reflux disease who were treated by Ljintang-Gamibang and acupuncture. The Journal of Internal Korean Medicine 2005, 26, 926-934.
- Lee, H.N.; Shin, S.A.; Choo, G.S.; Kim, H.J.; Park, Y.S.; Kim, B.S.; Kim, S.K.; Cho, S.D.; Nam, J.S.; Choi, C.S. Anti‑inflammatory effect of quercetin and galangin in LPS‑stimulated RAW264. 7 macrophages and DNCB‑induced atopic dermatitis animal models. International Journal of Molecular Medicine 2018, 41, 888-898.
- Byun, E.-B.; Yang, M.-S.; Choi, H.-G.; Sung, N.-Y.; Song, D.-S.; Sin, S.-J.; Byun, E.-H. Quercetin negatively regulates TLR4 signaling induced by lipopolysaccharide through Tollip expression. Biochemical and biophysical research communications 2013, 431, 698-705.
- Xu, J.; Li, Y.; Yang, X.; Li, H.; Xiao, X.; You, J.; Li, H.; Zheng, L.; Yi, C.; Li, Z. Quercetin inhibited LPS-induced cytokine storm by interacting with the AKT1-FoxO1 and Keap1-Nrf2 signaling pathway in macrophages. Scientific Reports 2024, 14, 20913.
- Wang, Z.; Okutsu, K.; Futagami, T.; Yoshizaki, Y.; Tamaki, H.; Maruyama, T.; Toume, K.; Komatsu, K.; Hashimoto, F.; Takamine, K. Microbial community structure and chemical constituents in Shinkiku, a fermented crude drug used in Kampo medicine. Frontiers in Nutrition 2020, 7, 115.
- Shan, L.; Yang, J.; Shi, Y.; Ye, Y.; Zhang, M.; Cui, Y.; Yu, H.; Wang, Y.; Chai, X. Metabolism and release of characteristic components and profiles of enzymatic activities during fermentation of Massa Medicata Fermentata. LWT 2024, 116793.
- Zhang, H.; Gao, S.; Zhang, X.; Meng, N.; Chai, X.; Wang, Y. Fermentation characteristics and the dynamic trend of chemical components during fermentation of Massa Medicata Fermentata. Arabian Journal of Chemistry 2022, 15, 103472.
- Tian, C.; Liu, X.; Chang, Y.; Wang, R.; Lv, T.; Cui, C.; Liu, M. Investigation of the anti-inflammatory and antioxidant activities of luteolin, kaempferol, apigenin and quercetin. South African Journal of Botany 2021, 137, 257-264.
- Moon, Y.J.; Wang, L.; DiCenzo, R.; Morris, M.E. Quercetin pharmacokinetics in humans. Biopharmaceutics & drug disposition 2008, 29, 205-217.
- Chen, X.; Yin, O.Q.; Zuo, Z.; Chow, M.S. Pharmacokinetics and modeling of quercetin and metabolites. Pharmaceutical research 2005, 22, 892-901.
- Orrego-Lagarón, N.; Martínez-Huélamo, M.; Quifer-Rada, P.; Lamuela-Raventos, R.M.; Escribano-Ferrer, E. Absorption and disposition of naringenin and quercetin after simultaneous administration via intestinal perfusion in mice. Food & function 2016, 7, 3880-3889.
- Zhang, C.; Jiang, D.; Wang, J.; Qi, Q. The effects of TPT and dietary quercetin on growth, hepatic oxidative damage and apoptosis in zebrafish. Ecotoxicology and Environmental Safety 2021, 224, 112697.
- Wang, J.; Zhang, C.; Zhang, J.; Xie, J.; Yang, L.; Xing, Y.; Li, Z. The effects of quercetin on immunity, antioxidant indices, and disease resistance in zebrafish (Danio rerio). Fish physiology and biochemistry 2020, 46, 759-770.
- Mani, R.J.; Mittal, K.; Katare, D.P. Protective effects of quercetin in zebrafish model of Alzheimer’s disease. Asian Journal of Pharmaceutics 2018, 12, S660.
Table References [CNKI references (written in Chinese)]
1 杜亚强,罗镭,陈碧莲.基于 UHPLC-LTQ-Orbitrap-MS/MS技术快速分析六神曲中的化学成分. 中国现代应用药学, 2024,41(16):2249-2256.DOI:10.13748/j.cnki.issn1007-7693.20230082.
2 李运,刘晓玲,许晓辉等. 固相萃取-超高效液相色谱-串联质谱法测定六神曲中4种烷基间苯二酚. 分析测试技术与仪器, 2024,30(04):251-259.DOI:10.16495/j.1006-3757.2024. 04.006.
3 丁海玲,时海燕,王爽等.六神曲发酵过程中酶学性质与微生物群落变化相关性研究. 中国医院药学杂志, 2024, 44(17):1980-1986.DOI:10.13286/j.1001-5213.2024.17.04.
4 张伟,杨直,金朦娜等. 不同发酵工艺对浙产六神曲中5种成分含量变化的影响. 中国医药导报, 2024, 21(10):16-19+42.DOI:10.20047/j.issn1673-7210.2024.10.04.
5 张希,朱月健,郑威等.六神曲成分的快速鉴定及脂类活性研究. 中国现代应用药学:1-12 [2024-10-09]. DOI:10.13748/j.cnki.issn1007-7693.20232227.
6 石亚玲,单鲁豫,杨晶晶等.基于定量核磁共振氢谱的神曲加工炮制过程中葡萄糖量变规律研究. 中国中药杂志,2023, 48(23):6396-6402.DOI:10.19540/j.cnki.cjcmm.20230919.301.
7 时海燕,徐男,赵霞等.基于HS-GC-IMS技术分析六神曲炮制前后(炒、焦)挥发性物质的变化. 中草药,2023, 54(10):3120-3131.
8 李虹霞,朱月健,尹磊等.六神曲化学成分及抗氧化活性研究. 中成药, 2023, 45(03):788-794.
9 高胜美,张欢,王跃飞等. 基于UHPLC-Q-Orbitrap技术的神曲发酵前后化学成分差异研究. 中成药, 2022, 44(12):3890-3895.

Round 2
Reviewer 1 Report
The authors well addressed my comments, and this work can be accepted.
The authors well addressed my comments.
Author Response
The authors well addressed my comments, and this work can be accepted.
- Thank you very much. Again, I am grateful for your valuable comments on our manuscript.
Reviewer 2 Report
The authors answer most objections, but at least part of the comments included in the accompanying letter should be adapted and included in the final version of the paper.
The authors answer most objections, but at least part of the comments included in the accompanying letter should be adapted and included in the final version of the paper.
Author Response
The authors answer most objections, but at least part of the comments included in the accompanying letter should be adapted and included in the final version of the paper.
- We appreciate for your sincere comments. We included major points of review discussion in revised manuscript with proper references. Details are below; Discussion part 1) As presented in the HPLC analysis, our MMF extract contains quercetin, which is equivalent to around 1.49 μM, which may be contributing to the mechanisms of systemic effects2) The standardization of MMF is indeed a significant issue for researchers. Consequently, there are many studies that investigated the major compounds of MMF, particularly following the fermentation process. Major composition profiles of MMFs from different batches have been investigated by researchers. A recent study investigated the differences among nine MMFs produced from various batches in China and Korea [65]. While it is not a simple issue, we may assert that many studies have been conducted and are currently underway by researchers to address the standardization issues of MMF. 3) We can reasonably assume that at least quercetin, as detected in our MMF sample, may help mitigate HCD or LPS-induced harmful conditions in zebrafish due to its notable antioxidant activities. We really appreciate for your valuable comments on our work. Thank you very much.